# Integration-free Training for Spatio-temporal Multimodal Covariate Deep Kernel Point Processes

**Yixuan Zhang**
China-Austria Belt and Road Joint Laboratory on
Artificial Intelligence and Advanced Manufacturing
Hangzhou Dianzi University
`yixuan.zhang@hdu.edu.cn`

**Quyu Kong**
Alibaba Group
`kongquyu.kqy@alibaba-inc.com`

**Feng Zhou** *
Center for Applied Statistics and School of Statistics
Renmin University of China
`feng.zhou@ruc.edu.cn`

## Abstract

In this study, we propose a novel deep spatio-temporal point process model, Deep Kernel Mixture Point Processes (DKMPP), that incorporates multimodal covariate information. DKMPP is an enhanced version of Deep Mixture Point Processes (DMPP), which uses a more flexible deep kernel to model complex relationships between events and covariate data, improving the model's expressiveness. To address the intractable training procedure of DKMPP due to the non-integrable deep kernel, we utilize an integration-free method based on score matching, and further improve efficiency by adopting a scalable denoising score matching method. Our experiments demonstrate that DKMPP and its corresponding score-based estimators outperform baseline models, showcasing the advantages of incorporating covariate information, utilizing a deep kernel, and employing score-based estimators.

## 1 Introduction

Point processes are widely used statistical tools for modeling event occurrence patterns within continuous spatio-temporal domains. They have garnered extensive attention across various fields, such as criminology [17; 38], neuroscience [24; 40], financial engineering [11; 2], and epidemiology [20; 25]. The core task of point processes is to derive an intensity function from event sequences, which represents the rate of event occurrences at any given time or location within the observation domain.

There are two common approaches for modeling intensity functions: traditional and covariate-based methods. The traditional approach [13; 10; 18] considers only event information, such as time and position, while ignoring the effects of contextual factors. Conversely, the covariate-based method [31; 33; 3] incorporates covariates from the observation domain, providing insights into factors that may cause event occurrences. In real applications, event occurrence rates are often correlated with contextual factors, making covariate-based methods more effective for event prediction [21]. For example, in crime data analysis, the covariate-based method considers factors like income level, education status, and public security to estimate crime intensity. Thus, utilizing a covariate-based point process model is significant for leveraging rich contextual information to predict events and explore various factors influencing event occurrence.

---

*Corresponding author.

37th Conference on Neural Information Processing Systems (NeurIPS 2023).

Okawa et al. [21] proposed Deep Mixture Point Processes (DMPP) to incorporate covariate information into point process models. DMPP models intensity as a deep mixture of parametric kernels, representing the influence of nearby representative points on the target event. Mixture weights are modeled by a neural network using covariates from representative points as input, avoiding intractable intensity integrals for log-likelihood evaluation and enabling simple parameter estimation. However, DMPP requires integrable parametric kernels like radial basis function (RBF) or polynomial kernels, which have closed-form solutions for intensity integrals. With more complex, flexible kernels (e.g., deep kernels [35]), DMPP cannot compute the intensity integral tractably. In real-world applications, flexible kernels are usually preferred to capture complex influences of covariates on intensity, which can be challenging for restricted parametric kernels.

In this study, we aim to learn the intricate influence of covariate data on event occurrence using a data-driven approach. We propose replacing the simple parametric kernel in DMPP with a deep kernel [35], which transforms the inputs of a base kernel using a deep neural network. The deep kernel significantly enhances expressive power by automatically learning a flexible input transformation metric, thus transcending Euclidean and absolute distance-based metrics. This results in a substantial improvement in DMPP's expressiveness, leading to our proposed model, Deep Kernel Mixture Point Processes (DKMPP). However, due to the deep architecture embedded in the base kernel, the deep kernel is non-integrable, causing an intractable training procedure as intensity integral evaluation is required for likelihood-based parameter estimation. To address parameter estimation in DKMPP, we adopt an integration-free method based on score matching [12], which estimates model parameters by matching the gradient of the log-density of the model and data. Score matching avoids intensity integral computation and allows tractable training. However, the naive score matching estimator is computationally expensive as it requires the computation of second derivatives of the log-density. To improve efficiency, we further adopt a scalable denoising score matching (DSM) method [32], with the resulting models referred to as score-DKMPP and score-DKMPP+ respectively.

Score-based estimators, such as Score-DKMPP and Score-DKMPP+, offer an advantage over traditional likelihood-based estimators as they do not require numerical integration, which is often computationally expensive and prone to errors. Additionally, Score-DKMPP+ has a high efficiency due to the utilization of denoising score matching. Our experiments serve to highlight the advantages of DKMPP and its corresponding score-based estimators. In the majority of our experiments, both Score-DKMPP and Score-DKMPP+ outperform the baseline models, indicating that incorporating covariate information can enhance the modeling and prediction of spatio-temporal events. Furthermore, our results strongly suggest that the adoption of a deep kernel significantly enhances the model's expressiveness when compared to a simple parametric kernel.

Specifically, our contributions are as follows: (1) We propose DKMPP, which replaces the simple parametric kernel in DMPP with a more flexible deep kernel, enhancing model expressiveness and allowing for the learning of complex influence from covariate factors on event occurrence. (2) We use the score matching estimator to overcome the intractable intensity integral in DKMPP during training and extend the denoising score matching method to DKMPP, significantly improving computational efficiency. (3) We analyze the difference between likelihood estimator and score matching estimators for DKMPP, demonstrate the superiority of DKMPP over baseline models, and show the impact of deep kernel and various hyperparameters through experiments.

## 2 Related Work

**Point processes** are statistical tools for modeling event occurrence patterns, with intensity functions characterizing event rates. Essential models include Poisson processes [13], renewal processes [5], and Hawkes processes [10]. Poisson processes feature independent point occurrences and have been widely used in economics [8], ecology [34], and astronomy [28]. Renewal processes generalize Poisson processes for arbitrary event intervals. However, neither model is suitable for self-excitation in event patterns. Hawkes processes, containing a triggering kernel, are suitable for applications with excitation effects, such as seismology [19], finance [11], criminology [16], and neuroscience [39].

**Deep point processes** have gained traction due to the growing popularity of deep neural networks. While traditional models capture simple event patterns, deep point processes leverage neural networks to describe complex dependencies. The first deep point process work by Du et al. [7] utilized RNNs to encode history information. Later, Xiao et al. [36] and Mei & Eisner [14] proposed LSTM-based

versions. However, RNNs struggle with long-term dependencies, training difficulties, and lack of parallelism. To address these issues, Zuo et al. [43] and Zhang et al. [37] proposed attention-based models. Additionally, there are some point process models directly relevant to our work, such as Okawa et al. [22] and Zhu et al. [42], which employ the use of deep kernels to model the intensity function. Besides directly modeling intensity functions with neural networks, other approaches use them to represent probability density functions [29] or cumulative intensity functions [23; 30]. However, to the best of our knowledge, most deep point process models are limited to temporal point processes, and few works have extended them to spatio-temporal point processes.

**Covariate point processes** differ from traditional point processes by incorporating contextual factors and establishing connections between these factors and event patterns. This integration of covariate information enables a more comprehensive explanation of studied phenomena and improves predictive performance. Meyer et al. [15] proposed a model for epidemic prediction using population density and other covariates. Adelfio & Chiodi [1] proposed a spatio-temporal self-exciting point process for earthquake forecasting using geological features. Gajardo & Müller [9] analyzed COVID-19 cases with point processes, studying their relation to covariates such as mobility restrictions and demographic factors. However, these works model the influence of covariate in a handcrafted parametric form, limiting flexibility.

Different from existing works, our approach emphasizes covariate point processes and enhances both model flexibility and parameter estimation. While our work is closely related to [21], we extend it by introducing the deep kernel and score-based estimation methods in our proposed DKMPP model.

## 3 Preliminaries

We offer a brief overview of three fundamental components that serve as the foundation for our work.

### 3.1 Spatio-temporal Point Processes

A spatio-temporal point process is a stochastic model whose realization is a random collection of points representing the time and location of events [6]. Consider a sequence of $N$ events $S = \{(t_1, \mathbf{x}_1), (t_2, \mathbf{x}_2), \ldots, (t_N, \mathbf{x}_N)\}$, where $(t_n, \mathbf{x}_n)$ denotes the time and location of the $n$-th event in $\mathcal{T} \times \mathcal{X} \subseteq \mathcal{R} \times \mathcal{R}^2$. The intensity function $\lambda(t, \mathbf{x})$ represents the instantaneous event occurrence rate:

$$\lambda(t, \mathbf{x}) = \lim_{|\Delta_t|, |\Delta_\mathbf{x}| \to 0} \frac{\mathbb{E}[\mathbb{N}(\Delta_t \times \Delta_\mathbf{x})]}{|\Delta_t||\Delta_\mathbf{x}|}, \tag{1}$$

where $\mathbb{N}(\Delta_t \times \Delta_\mathbf{x})$ is the number of events in $\Delta_t \times \Delta_\mathbf{x}$, $\Delta_t$ is a small time interval around $t$, $\Delta_\mathbf{x}$ is a small spatial region around $\mathbf{x}$, and $\mathbb{E}$ denotes expectation w.r.t. realizations. Given a sequence $S$, the probability density function, or likelihood, of the point process can be written as [6]:

$$p(S \mid \lambda(t, \mathbf{x})) = \prod_{n=1}^{N} \lambda(t_n, \mathbf{x}_n) \exp\left(-\int_{\mathcal{T} \times \mathcal{X}} \lambda(t, \mathbf{x}) dt d\mathbf{x}\right). \tag{2}$$

### 3.2 Deep Kernel

Wilson et al. [35] combined kernel methods and neural networks to create a deep kernel, capturing the expressive power of deep neural networks. The deep kernel extends traditional covariance kernels by embedding a deep architecture into the base kernel:

$$k_\phi(\mathbf{x}, \mathbf{x}') \to k_\phi(g_w(\mathbf{x}), g_w(\mathbf{x}')), \tag{3}$$

where $k_\phi$ is a base kernel with parameters $\phi$. To enhance flexibility, the inputs $x$ and $x'$ are transformed by a deep neural network $g$ parameterized by weights $w$. The base kernel offers various options, such as the common RBF kernel. The deep kernel parameters include base kernel parameters $\phi$ and neural network weights $w$. An advantage of deep kernels is their ability to learn metrics by optimizing input space transformation in a data-driven manner, rather than relying on Euclidean distance-based metrics, which are common in traditional kernels but may not always be suitable, especially in high-dimensional input spaces.

### 3.3 Score Matching

The classic maximum likelihood estimation (MLE) minimizes the Kullback–Leibler (KL) divergence between the parameterized model distribution and the data distribution, but requires an often intractable normalizing constant. Numerical integration can approximate the intractable integral, but its complexity grows exponentially with the input dimension. An alternative is to minimize the Fisher divergence, which does not require the normalizing constant. The Fisher divergence is defined as:

$$F(p(x), q(x)) = \frac{1}{2}\mathbb{E}_{p(x)}\|\nabla \log p(x) - \nabla \log q(x)\|^2, \tag{4}$$

where $p(x)$ and $q(x)$ are two smooth distributions of the same dimension, and $\|\cdot\|$ is a suitable norm (e.g., $\ell^2$ norm). Minimizing the Fisher divergence estimates model parameters by matching the gradient of the log-density of the model to the log-density of the data. The parameterized model density is defined as $p_\theta(x) = \frac{1}{Z(\theta)}\tilde{p}(x \mid \theta)$, where $\theta$ represents the model parameters, and the normalizing constant $Z(\theta) = \int \tilde{p}(x \mid \theta)dx$ is intractable. With Fisher divergence, we do not need to compute the normalizing constant, as the matched gradients do not depend on it. The gradient of log-density is referred to score, so the aforementioned method is also called score matching [12].

## 4 Methodology

In this section, we introduce our proposed DKMPP and two score-based parameter estimation methods. Following Okawa et al. [21], we use the same notation. We assume a sequence of $N$ events $S = \{\mathbf{s}_1 = (t_1, \mathbf{x}_1), \mathbf{s}_2 = (t_2, \mathbf{x}_2), \ldots, \mathbf{s}_N = (t_N, \mathbf{x}_N)\}$, where $\mathbf{s}_n = (t_n, \mathbf{x}_n)$ represents the time and location of the $n$-th event in $\mathcal{T} \times \mathcal{X} \subseteq \mathcal{R} \times \mathcal{R}^2$. We also assume $K$ contextual covariates on the spatio-temporal region $\mathcal{T} \times \mathcal{X}$: $\mathcal{D} = \{Z_1, Z_2, \ldots, Z_K\}$, where $Z_k$ is the $k$-th covariate.

### 4.1 Deep Kernel Mixture Point Processes

Okawa et al. [21] developed a deep spatio-temporal covariate point process model called DMPP, capable of incorporating multimodal contextual data like temperature, humidity, text and image. The intensity of DMPP is designed in a kernel convolution form:

$$\lambda(\mathbf{s} \mid \mathcal{D}) = \int f_w(\mathbf{u}, \mathbf{Z}(\mathbf{u}))k_\phi(\mathbf{s}, \mathbf{u})d\mathbf{u}, \tag{5}$$

where $\mathbf{u} = (\tau, \mathbf{r})$ is a point in the region $\mathcal{T} \times \mathcal{X}$, and $k_\phi(\mathbf{s}, \mathbf{u})$ is a kernel with parameters $\phi$. $f_w$, a deep neural network, inputs contextual data and outputs nonnegative mixture weights, with $w$ representing the network parameters. $\mathbf{Z}(\mathbf{u}) = \{Z_1(\mathbf{u}), \ldots, Z_K(\mathbf{u})\}$ are covariate values at point $\mathbf{u}$.

Although the intensity formulation integrates multimodal contextual data, in reality, covariate data is only available on a regular grid. DMPP introduces finite representative points $\{\mathbf{u}_j\}_{j=1}^J$ on the spatio-temporal domain, resulting in a discrete version[2]:

$$\lambda(\mathbf{s} \mid \mathcal{D}) = \sum_{j=1}^J f_w(\mathbf{u}_j, \mathbf{Z}(\mathbf{u}_j))k_\phi(\mathbf{s}, \mathbf{u}_j). \tag{6}$$

The discrete version of DMPP offers two significant advantages: (1) the intensity itself is tractable since it requires summation instead of integration; (2) the intensity integral is tractable as long as the kernel integral is tractable, as given by $\int \lambda(\mathbf{s} \mid \mathcal{D})d\mathbf{s} = \sum_{j=1}^J f_w(\mathbf{u}_j, \mathbf{Z}(\mathbf{u}_j)) \int k_\phi(\mathbf{s}, \mathbf{u}_j)d\mathbf{s}$. For this reason, DMPP employs simple integrable kernels like RBF kernels, which can be integrated analytically, to simplify parameter estimation. This design choice ensures that the intensity integral is tractable, which is crucial for evaluating log-likelihood. Consequently, parameter estimation can be performed using straightforward back-propagation.

However, this approach has notable drawbacks. Firstly, using limited parametric kernels constrains the intensity's expressive power. Secondly, Euclidean distance in the kernel may not be a suitable measure of similarity, particularly in high-dimensional input spaces. In many applications, the

---

[2]The intensities in Eqs. (6) and (7) are approximations of the true intensity in Eq. (5). Since we only use the approximation in the following, we adopt the same notation without explicitly distinguishing between them.

relationship between covariates and event occurrence is complex and often unknown, making it desirable to model this influence using data-driven kernels with non-Euclidean distance metrics.

To address the issues mentioned earlier, we propose DKMPP, which replaces the simple integrable kernel in DMPP with the deep kernel [35]. The deep kernel captures the similarity between representative points and event points in both time and location. The deep kernel has greater expressiveness than the traditional kernel and can flexibly learn the metric's functional form through a neural network-based nonlinear transformation. Specifically, the intensity of DKMPP is designed as:

$$\lambda(\mathbf{s} \mid \mathcal{D}) = \eta \left( \sum_{j=1}^{J} f_{w_1}(\mathbf{u}_j, \mathbf{Z}(\mathbf{u}_j)) k_{\phi, w_2}(\mathbf{s}, \mathbf{u}_j) \right), \tag{7}$$

where $k_{\phi, w_2}(\mathbf{x}, \mathbf{x}') = k_\phi(g_{w_2}(\mathbf{x}), g_{w_2}(\mathbf{x}'))$ is the deep kernel, with $\phi$ as the base kernel parameters and $w_2$ as the weights of a nonlinear transformation $g$ implemented by a deep neural network. In subsequent experiments, we found that, even for this three-dimensional problem, the deep kernel's ability to learn metrics from data outperforms the traditional Euclidean distance. Compared to DMPP, DKMPP introduces a link function $\eta(\cdot)$ to ensure non-negativity of the intensity function[3]. Various functions, such as softplus, exponential, and ReLU, can be utilized as the link function. Unless otherwise specified, we choose softplus as $\eta(\cdot)$ in this work.

**Representative Points**    In this study, we adopt a fixed grid of representative points, which are evenly spaced both along the temporal axis and across the spatial region. The complete set of representative points is derived as the Cartesian product of the temporal and spatial representatives.

**Implementation Description**    Both the kernel mixture weight network $f$ and the non-linear transformation $g$ in the deep kernel are implemented using MLPs with ReLU activation functions. To handle multimodal covariates such as numerical values, categorical values, and text, etc., we perform necessary feature extraction. If the covariates are numerical, we normalize them as necessary. If the covariates are categorical, we convert them to numerical type or one-hot encoding. If the covariates are textual, we use a Transformer architecture to extract features. If the covariates are multimodal, we fuse the multimodal information by concatenating the extracted features from multiple modules.

### 4.2    Parameter Estimation with Score Matching

The traditional approach to estimate point process is based on MLE where the log-likelihood is:

$$\log p_\theta(S) = \log p(\{\mathbf{s}_n\}_{n=1}^{N} \mid \lambda_\theta(\mathbf{s})) = \sum_{n=1}^{N} \log \lambda_\theta(\mathbf{s}_n) - \int_{\mathcal{T} \times \mathcal{X}} \lambda_\theta(\mathbf{s}) d\mathbf{s}. \tag{8}$$

The intensity integral is a compensator which can be understood as a normalizing constant. The integration is intractable in most cases, and our proposed DKMPP is no exception. To solve this problem, we normally resort to numerical integration, e.g., Monte Carlo or quadrature, to obtain an approximation. However, these numerical methods introduce additional errors; more importantly, they are not scalable to the high-dimensional problem.

**Score-DKMPP**  In this work, we propose an integration-free estimation method based on score matching [12] to estimate the model parameters in our proposed DKMPP. Surprisingly, the score-based estimator for point process model is largely unexplored in recent years. As far as we know, few works attempted to utilize score matching for the estimation of point processes. Sahani et al. [26] proposed a pioneering estimator in this field, linking score matching with temporal point processes. Here, we extend the score matching estimator to our covariate-based spatio-temporal DKMPP. We assume the ground-truth process generating the data has a density $p(S)$ and design a parameterized model with density $p_\theta(S)$ where $\theta$ is the model parameter to estimate. A Fisher-divergence objective for our covariate-based spatio-temporal DKMPP is designed as:

$$F(\theta) = \mathbb{E}_{p(S)} \frac{1}{2} \sum_{n=1}^{\tilde{N}} \left( \frac{\partial \log p(S)}{\partial s_n} - \frac{\partial \log p_\theta(S)}{\partial s_n} \right)^2, \tag{9}$$

---

[3]In DMPP (Eq. (6)), $f_w$ only produces non-negative weights. However, in DKMPP (Eq. (7)), we remove the non-negative constraint of $f_w$ and introduce a link function to ensure the non-negativity of the intensity.

where $s_n$ is any entry in vector $\mathbf{s}_n$, i.e., $s_n \in \mathbf{s}_n = (t_n, \mathbf{x}_n)$, $\tilde{N}$ is the number of equivalent variables, $\tilde{N} = N$ for 1-D point process, e.g., temporal point process; $\tilde{N} = 2N$ for 2-D point process, e.g., spatial point process; and $\tilde{N} = 3N$ for 3-D point process, e.g., spatio-temporal point process, etc. The optimal estimate of model parameter is given by $\hat{\theta} = \arg\min_\theta F(\theta)$.

However, the above loss cannot be minimized directly as it depends on the gradient of the ground-truth data distribution which is unknown. Following the derivation in Hyvärinen [12], this dependence can be eliminated by using a trick of integration by parts. Sahani et al. [26] assumed the log density of point process satisfies the proper smoothness and obtained a concise empirical loss. In this work, we prove that without those smoothness assumptions, we can still obtain the same empirical loss (proof provided in Appendix A):

$$\hat{F}(\theta) = \frac{1}{M} \sum_{m=1}^{M} \sum_{n=1}^{\tilde{N}_m} \frac{1}{2} \left( \frac{\partial \log p_\theta(S_m)}{\partial s_{m,n}} \right)^2 + \frac{\partial^2 \log p_\theta(S_m)}{\partial s_{m,n}^2} + C_1, \tag{10}$$

where we take $M$ sequences $\{S_m\}_{m=1}^{M}$ from $p(S)$, $\tilde{N}_m$ is the number of equivalent variables on the $m$-th sequence, $s_{m,n}$ is the $n$-th variable on the $m$-th sequence, the constant $C_1$ does not depend on $\theta$ and can be discarded. It is easy to see that, in Eq. (10), the intensity integral has been removed by the operation of gradient. We refer to our proposed DKMPP estimated by Eq. (10) as score-DKMPP.

**Score-DKMPP+** A serious drawback of the objective in Eq. (10) is it requires the second derivative which is computationally expensive. To improve efficiency, we derive the denoising score matching (DSM) method [32] for our proposed DKMPP to avoid second derivatives. We add a small noise to the sequence $S$ to obtain a noisy sequence $\tilde{S}$ (we add noise to each variable $\tilde{s}_n = s_n + \epsilon$), which is distributed as $p(\tilde{S}) = \int p(\tilde{S} \mid S)p(S)dS$. The DSM method uses the Fisher divergence between the noisy data distribution $p(\tilde{S})$ and model distribution $p_\theta(\tilde{S})$ as an objective:

$$F_{\text{DSM}}(\theta) = \mathbb{E}_{p(\tilde{s})} \frac{1}{2} \sum_{n=1}^{\tilde{N}} \left( \frac{\partial \log p(\tilde{S})}{\partial \tilde{s}_n} - \frac{\partial \log p_\theta(\tilde{S})}{\partial \tilde{s}_n} \right)^2, \tag{11}$$

where $\tilde{s}_n$ is any entry in the noisy vector $\tilde{\mathbf{s}}_n$, i.e., $\tilde{s}_n \in \tilde{\mathbf{s}}_n = (\tilde{t}_n, \tilde{\mathbf{x}}_n)$. The optimal parameter estimate is given by $\hat{\theta} = \arg\min_\theta F_{\text{DSM}}(\theta)$. To facilitate computation, the Fisher divergence in Eq. (11) can be further rewritten as (proof provided in Appendix B):

$$\hat{F}_{\text{DSM}}(\theta) = \frac{1}{2M} \sum_{m=1}^{M} \sum_{n=1}^{\tilde{N}_m} \left( \frac{\partial \log p(\tilde{S}_m \mid S_m)}{\partial \tilde{s}_{m,n}} - \frac{\partial \log p_\theta(\tilde{S}_m)}{\partial \tilde{s}_{m,n}} \right)^2 + C_2, \tag{12}$$

where we take $M$ clean and noisy sequences $\{S_m, \tilde{S}_m\}_{m=1}^{M}$ from $p(S, \tilde{S})$, $\tilde{s}_{m,n}$ is the $n$-th variable on the $m$-th noisy sequence, the constant $C_2$ does not depend on $\theta$ and can be discarded. With a Gaussian noise $\epsilon \sim \mathcal{N}(0, \sigma^2)$, the conditional gradient term can be written in a closed form: $\partial \log p(\tilde{S}_m \mid S_m)/\partial \tilde{s}_{m,n} = -(\tilde{s}_{m,n} - s_{m,n})/\sigma^2$. Equation (12) avoids the unknown ground-truth data distribution, the intractable intensity integral and the tedious second derivatives.

It is worth noting that the denoising score matching is not equivalent to the original score matching, because it is easy to see from Eq. (11) that the model is trained to match the score of the noisy data distribution $p(\tilde{S})$ instead of the original $p(S)$. However, when the noise is small enough, we can consider $p(\tilde{S}) \approx p(S)$ such that Eq. (12) is approximately equivalent to Eq. (10). We refer to our proposed DKMPP estimated by Eq. (12) as score-DKMPP+.

## 5 Experiments

In the experimental section, we mainly analyze the difference between MLE and score matching for DKMPP, the improvement in performance of DKMPP over baseline models, as well as the impact of various hyperparameters.

### 5.1 DKMPP: MLE vs. Score Matching

Since our proposed DKMPP model employs a score-based parameter estimation method instead of traditional MLE, a natural question arises regarding the comparison between the two approaches. To

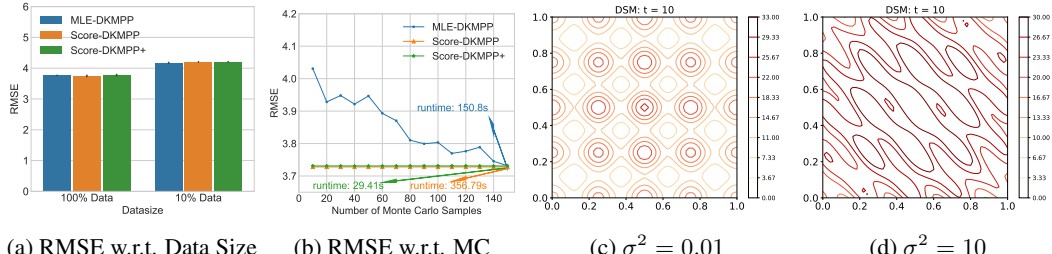

| (a) RMSE w.r.t. Data Size | (b) RMSE w.r.t. MC | (c) $\sigma^2 = 0.01$ | (d) $\sigma^2 = 10$ |

Figure 1: (a) The RMSE performance of MLE-DKMPP, Score-DKMPP and Score-DKMPP+ with $100\%$ and $10\%$ training data (the number of Monte Carlo samples is fixed to 1,000); (b) the RMSE performance of three estimators with the number of Monte Carlo samples ranging from 10 to 150; (c) the learned intensity function at $t = 10$ from Score-DKMPP+ with the noise variance $\sigma^2 = 0.01$; (d) the learned intensity function at $t = 10$ from Score-DKMPP+ with $\sigma^2 = 10$.

evaluate this question, we generate a 3-D spatio-temporal point process synthetic dataset. The spatial observation $\mathbf{x}$ spans the area of $[0, 1] \times [0, 1]$, while the temporal observation window covers the time interval of $[0, 10]$. We design a 1-D covariate function on the domain, a kernel mixture weight function $f_{w_1}(\mathbf{u}_j, Z(\mathbf{u}_j))$, and a deep kernel $k_{\phi,w_2}(\mathbf{s}, \mathbf{u}_j)$ with the RBF base kernel. More details are provided in Appendix C.1. We fix the representative points on a regular grid: 5 representative points evenly spaced on each axis, so there are $5^3 = 125$ representative points in total. We use the thinning algorithm [19] to generate 5,000 sequences according to the ground truth specified above. The statistics of the synthetic data are provided in Appendix C.1. We try to fit a DKMPP model to the synthetic data with the ground-truth representative points and an RBF base kernel. Both the kernel mixture weight network $f$ and the non-linear transformation $g$ in the deep kernel are implemented using MLPs with ReLU activation functions. Therefore, the learnable parameters are $w_1, w_2, \phi$.

Two aspects of particular interest when comparing MLE and score matching estimators are consistency and computational cost. Consistency refers to the property that as the data size varies, the estimates produced by these methods converge to the true parameter values. The computational costs of these methods refer to the amount of time required to achieve a certain level of accuracy.

For consistency, we use the root mean square error (RMSE) between the estimated intensity and the ground-truth intensity as a metric. During the training of MLE, numerical integration is required to approximate the intractable intensity integral. In our experiments, we use Monte Carlo methods and fix the number of Monte Carlo samples to 1,000, which is sufficient to obtain good results. For Score-DKMPP+, we use a Gaussian noise $\epsilon \sim \mathcal{N}(0, \sigma^2)$ with $\sigma^2 = 0.01$. Refer to Appendix C.1 for the intensity functions learned by different estimators. We test the RMSE of MLE-DKMPP, Score-DKMPP and Score-DKMPP+ under different data sizes ($100\%$ and $10\%$), as shown in Fig. 1a. Three estimators exhibit very similar RMSE performance; however, as the data size decreases, the errors of all three estimators increase slightly. Besides, we examine the effect of Monte Carlo samples on RMSE. Specifically, we use $100\%$ data and increase the number of Monte Carlo samples from 10 to 150. The results are shown in Fig. 1b. Due to the reliance on Monte Carlo integration, MLE suffers from poor RMSE when the number of Monte Carlo samples is insufficient. As the number of Monte Carlo samples increases, the parameter estimation performance of MLE improves and eventually converges. In comparison, score matching estimators (Score-DKMPP and Score-DKMPP+) have an advantage because they only require event location and do not rely on Monte Carlo integration. Therefore, the parameter estimation performance of score matching estimators is independent of the number of Monte Carlo samples.

For computational efficiency, we record the running time required by three estimators to achieve the same level of accuracy. The results are shown in Fig. 1b, indicating that to reach the same RMSE, the Score-DKMPP requires 356.8 seconds, while the MLE is faster, taking 150.8 seconds. Furthermore, the Score-DKMPP+ is extremely fast, completing the task in only 29.4 seconds.

## 5.2 Real-world Data

In this section, we validate our proposed DKMPP model and its corresponding score-based estimators on real-world data from various domains. We also compare DKMPP against other popular spatial-temporal point process models.

**Datasets** We analyze three datasets from the fields of transportation and crime, with details of the datasets shown below. Each dataset is divided into training, validation and test data using a $50\%/40\%/10\%$ split ratio based on time. The data preprocessing is provided in Appendix C.2.

*Crimes in Vancouver*[4] This dataset is composed of more than 530 thousand crime records, including all categories of crimes committed in Vancouver. Each crime record contains the time and location (latitude and longitude) of the crime.

*NYC Vehicle Collisions*[5] The New York City vehicle collision dataset contains about 1.05 million vehicle collision records. Each collision record includes the time and location (latitude and longitude).

*NYC Complaint Data*[6] This dataset contains over 228 thousand complaint records in New York City. Each record includes the date, time, and location (latitude and longitude) of the complaint.

*Covariates* These three datasets contain not only the spatio-temporal information of the events but also multimodal covariate information. For example, the Crime in Vancouver dataset includes information on the cause and type of crime records; the NYC Vehicle Collisions dataset includes textual descriptions of the accident scene, such as borough and contributing factors; the NYC Complaint Data includes numerical and textual covariate information, such as the police department number and offense description. For every representative point, we identify the closest numerical/categorical/textual features in both time and space, which are used as covariates.

**Baselines** Deep temporal point process models have gained widespread use in recent years, e.g., Mei & Eisner [14]; Omi et al. [23]; Zhang et al. [37]; Zuo et al. [43]. These works primarily focus on modeling time information while neglecting spatial information. As far as we know, there has been limited research on deep learning approaches for spatial-temporal point process modeling. Here, for a fair comparison, we compare DKMPP against deep (covariate) spatial-temporal point process models. Specifically, the following baselines are considered: (1) the homogeneous Poisson process (HomoPoisson) [13]; (2) the neural spatio-temporal point processes (NSTPP) [4]; (3) the deep spatio-temporal point processes (DeepSTPP) [41]; (4) the DMPP [21].

HomoPoisson is a traditional statistical spatial-temporal point process model; NSTPP and DeepSTPP are deep spatial-temporal point process models that do not incorporate covariates, while DMPP and DKMPP are deep covariate spatial-temporal point process models that allow the inclusion of covariate information. For DMPP and DKMPP, we experiment with three different kernels: RBF kernel, rational quadratic (RQ) kernel and Ornstein-Uhlenbeck (OU) kernel (See Appendix C.2).

**Metrics** We use two metrics to evaluate the performance: test log-likelihood (TLL) and prediction accuracy (ACC). The TLL measures the log-likelihood on the test data, indicating how well the model captures the distribution of the data. The ACC evaluates the absolute difference between the predicted number of events and the actual number of events on the test data, $1 - |\text{predicted \#} - \text{actual \#}|/\text{actual \#}$, assessing how accurately the model predicts the number of events on the test data.

**Results** We evaluate the performance of all baseline models in terms of TLL and ACC. In our experiments, we use Monte Carlo methods for the baselines which need numerical integration and fix the number of Monte Carlo samples to 1,000 which is sufficient to obtain good results. For Score-DKMPP+, we use a Gaussian noise $\epsilon \sim \mathcal{N}(0, \sigma^2)$ with $\sigma^2 = 0.01$. More training details are provided in Appendix C.2. The performance results are presented in Table 1, which show that Score-DKMPP and Score-DKMPP+ achieve similar performance with the same kernel, and both outperform other baseline models. As expected, HomoPoisson performs the worst due to its lack of flexibility in modeling varying intensity functions over time and space. In comparison to NSTPP and DeepSTPP, DKMPP performs better in two aspects. Firstly, DKMPP incorporates rich multimodal covariate information, which aids in modeling event occurrences. Secondly, DKMPP employs a score-based estimator instead of the MLE estimator, which avoids unnecessary errors caused by numerical integration. The comparison between DKMPP and DMPP with the same kernel constitutes an ablation study, which again verifies that using a deep kernel can provide stronger representational power and achieve better performance in both data fitting and prediction. This is because the deep

---

[4]https://www.kaggle.com/datasets/wosaku/crime-in-vancouver
[5]https://data.cityofnewyork.us/Public-Safety/NYPD-Motor-Vehicle-Collisions/h9gi-nx95
[6]https://data.cityofnewyork.us/Public-Safety/NYPD-Complaint-Data-Current-YTD/5uac-w243

Table 1: The performance of TLL and ACC (mean±std) for DKMPP and baseline models on three real-world datasets. For DMPP and DKMPP, we experiment with RBF kernel, RQ kernel and OU kernel. Both for TLL and ACC, higher values indicate better performance.

| Model | Crimes in Vancouver | | NYC Vehicle Collisions | | NYC Complaint Data | |
|---|---|---|---|---|---|---|
| | TLL | ACC (%) | TLL | ACC (%) | TLL | ACC (%) |
| HomoPoisson | 60.468 | 52.516 | 394.890 | 66.811 | 1.660 | 26.834 |
| NSTPP | 65.45±4.46 | 74.22±1.36 | 396.33±19.66 | 72.08±3.53 | 3.07±0.38 | 32.83±1.41 |
| DeepSTPP | 62.15±6.16 | 69.89±4.26 | 396.60±11.28 | 69.35±2.07 | 3.97±1.38 | 43.72±0.96 |
| DMPP (RBF) | 62.58±1.02 | 56.90±0.98 | 394.54±2.17 | 67.91±1.55 | 2.06±0.56 | 30.97±1.23 |
| DMPP (RQ) | 60.09±0.97 | 51.52±1.01 | 399.71±1.99 | 72.32±1.23 | 2.11±0.32 | 31.92±0.68 |
| DMPP (OU) | 60.85±1.13 | 52.59±0.96 | 402.54±2.49 | 74.27±1.03 | 2.02±0.21 | 29.95±0.57 |
| Score-DKMPP (RBF) | 69.06±0.68 | 78.71±0.37 | 409.02±0.92 | 79.67±0.65 | 2.96±0.11 | 43.40±0.78 |
| Score-DKMPP (RQ) | **69.55±0.39** | 80.16±1.01 | **411.03±1.12** | 79.42±0.79 | 3.04±0.23 | 44.57±0.26 |
| Score-DKMPP (OU) | 69.51±0.49 | 78.41±0.83 | 407.43±1.24 | 78.84±0.87 | 3.06±0.16 | 44.62±1.39 |
| Score-DKMPP+ (RBF) | 67.03±0.23 | **80.20±0.34** | 402.54±1.06 | 79.13±0.67 | 3.13±0.28 | 46.48±0.43 |
| Score-DKMPP+ (RQ) | 69.52±1.14 | 80.09±0.90 | 403.93±1.43 | 79.85±1.89 | 3.74±0.35 | 47.34±1.44 |
| Score-DKMPP+ (OU) | 68.78±0.90 | 80.03±0.69 | 400.32±1.16 | **79.86±1.08** | **4.28±0.11** | **47.36±0.79** |

Table 2: The RMSE performance (mean±std) of Score-DKMPP and Score-DKMPP+ on the synthetic dataset with various combinations of hyperparameters: for the representative points, we used 1-layer MLPs and a batch size of 100; for the number of layers, we used 125 representative points and a batch size of 100; and for the batch size, we used 125 representative points and 1-layer MLPs.

| Model/RMSE | Representative Points | | | Number of Layers | | | Batch Size | | |
|---|---|---|---|---|---|---|---|---|---|
| | 64 | 125 | 216 | 1 | 2 | 4 | 50 | 100 | 200 |
| Score-DKMPP | 3.86±0.26 | **3.66±0.27** | **3.66±0.26** | **3.84±0.23** | 4.00±0.21 | 4.13±0.16 | 3.60±0.35 | **3.47±0.23** | 4.07±0.17 |
| Score-DKMPP+ | 3.78±0.23 | 3.73±0.21 | **3.72±0.22** | **3.78±0.17** | 3.95±0.23 | 4.08±0.26 | 3.67±0.31 | **3.61±0.21** | 3.65±0.15 |

kernel can flexibly learn the functional form of the metric by optimizing input space transformation in a data-driven manner, rather than relying on a fixed Euclidean distance-based metric.

## 5.3 Hyperparameter Analysis

**Representative Points** The effect of the number of representative points on the performance of DKMPP on synthetic and real-world data is presented in Table 2 and Appendix C.2, respectively. In general, the accuracy of DKMPP tends to improve with an increase in the number of representative points, although the change in performance is not significant. However, having more representative points leads to higher computational costs. Therefore, in the experiment, the number of representative points needs to be chosen as a trade-off between accuracy and efficiency.

**Layer and Batch Size** We examine how the number of layers in the kernel mixture weight network $f$ and the non-linear transformation $g$ of the deep kernel, as well as batch size, affect model performance on synthetic and real data in Table 2 and Appendix C.2, respectively. In general, varying batch sizes do not produce a significant difference in model performance, so we set a batch size of 100 because both overly large and excessively small batch sizes may harm optimization. For the number of layers, we find that the number of layers is correlated with the dimension of covariates. In the case of low-dimensional covariates, a simple hidden layer can provide better performance than multiple layers. Thus, we set the number of layers to 1 for the synthetic data and 2 for the real data.

**Noise Variance** The noise added by the Score-DKMPP+ is an important hyperparameter. If the noise is too large, the resulting noisy distribution $p(\tilde{S})$ will deviate significantly from the true data distribution $p(S)$, leading to severe biases in the learned results. We conduct experiments with two different noise variance settings, $\sigma^2 = 0.01$ and $\sigma^2 = 10$. When $\sigma^2 = 0.01$, the model can learn the intensity function that is close to the ground truth, as shown in Fig. 1c. However, as the noise level increases, when $\sigma^2 = 10$, the learned results deviate significantly from the true values, as shown in Fig. 1d. In practice, setting $\sigma^2$ too small can also lead to numerical instability.

## 6 Limitations

One limitation of the current study is that the proposed multimodal covariate spatio-temporal point process model is constrained to covariates of numerical, categorical, and textual types during the experiments, primarily due to the availability of data. However, future research could greatly benefit from the incorporation of image-type covariates, as this holds immense potential for enhancing the model's ability to capture a wider range of diverse and intricate effects. By including image data, the model could potentially uncover richer patterns and relationships, leading to even more fascinating and insightful results.

## 7 Conlusions

In conclusion, we have introduced a novel deep spatio-temporal point process model, the DKMPP, which incorporates multimodal covariate information and captures complex relationships between events and covariate data by utilizing a more flexible deep kernel leading to improved expressiveness. We address the intractable training procedure of DKMPP by using an integration-free method based on score matching and improving efficiency through a scalable denoising score matching method. Through our experiments, we show that DKMPP and its corresponding score-based estimators outperform baseline models, demonstrating the advantages of incorporating covariate information, utilizing a deep kernel, and employing score-based estimators. Our results suggest that DKMPP is a promising approach for modeling complex spatio-temporal events with multimodal covariate data.

### Acknowledgments and Disclosure of Funding

This work was supported by NSFC Project (No. 62106121), the MOE Project of Key Research Institute of Humanities and Social Sciences (22JJD110001), the fund for building world-class universities (disciplines) of Renmin University of China (No. KYGJC2023012), China-Austria Belt and Road Joint Laboratory on Artificial Intelligence and Advanced Manufacturing of Hangzhou Dianzi University, and the Public Computing Cloud, Renmin University of China.

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

# A Score Matching Estimator

We provide the proof of the score matching estimator for temporal point processes and spatial point processes below, respectively. Generally speaking, the derivation for temporal point processes is more complex than spatial point processes because as we can see later there exists some complications arising from different limits of integration due to the order constraint on timestamps.

## A.1 Temporal Point Processes

Given an observation window $[0, T]$, a sequence from a temporal point process is composed of a random number of timestamps arranged in a sequential order $S = \{t_n\}_{n=1}^{N}$ where $t_1 < t_2 < \ldots < t_N$ and $t_n \in [0, T]$ is the $n$-th event timestamp. We assume the ground-truth process generating the data has a density $p(S)$ and design a parameterized model with density $p_\theta(S)$ where $\theta$ is the model parameter to estimate. Following Sahani et al. [26], we define a Fisher-divergence objective:

$$F(\theta) = \mathbb{E}_{p(S)} \frac{1}{2} \sum_{n=1}^{N} \left( \frac{\partial \log p(S)}{\partial t_n} - \frac{\partial \log p_\theta(S)}{\partial t_n} \right)^2. \tag{13}$$

The above loss can be understood as matching the variational derivatives of log-density w.r.t. the counting process $N(t)$ given that the counting process is non-decreasing and piece-wise constant with unit steps.

However, the above loss cannot be minimized directly as it depends on the gradient of the ground-truth data distribution which is unknown. Following the derivation of Hyvärinen [12], this dependence can be eliminated by using a trick of integration by parts. Let us expand Eq. (13), discard $(\frac{\partial \log p(S)}{\partial t_n})^2$ which does not depend on parameter $\theta$, and examine the cross-term:

$$\mathbb{E}_{p(S)} \left[ \frac{\partial \log p(S)}{\partial t_n} \frac{\partial \log p_\theta(S)}{\partial t_n} \right]$$

$$= \int p(S) \frac{\partial \log p(S)}{\partial t_n} \frac{\partial \log p_\theta(S)}{\partial t_n} dS$$

$$= \int_{S_{t_n^-}} \int_{t_n} \frac{\partial p(S)}{\partial t_n} \frac{\partial \log p_\theta(S)}{\partial t_n} dt_n dS_{t_n^-} \tag{14}$$

$$= \int_{S_{t_n^-}} p(S) \frac{\partial \log p_\theta(S)}{\partial t_n} \Big|_{t_n = t_{n-1}}^{t_n = t_{n+1}} - \int_{t_n} p(S) \frac{\partial^2 \log p_\theta(S)}{\partial t_n^2} dt_n dS_{t_n^-}$$

$$= \mathbb{E}_{p(S)} \left[ \frac{\partial \log p_\theta(S)}{\partial t_n} (\delta(t_n - t_{n+1}) - \delta(t_n - t_{n-1})) - \frac{\partial^2 \log p_\theta(S)}{\partial t_n^2} \right].$$

where $S_{t_n^-}$ represents the sequence excluding $t_n$, the fourth line uses integration by parts, the fifth line uses delta function to evaluate the limits of $t_n$ given the order constraint of timestamps. Therefore, the loss in Eq. (13) can be rewritten as:

$$F(\theta) = \mathbb{E}_{p(S)} \left[ \sum_{n=1}^{N} \frac{1}{2} \left( \frac{\partial \log p_\theta(S)}{\partial t_n} \right)^2 \right.$$

$$\left. - \frac{\partial \log p_\theta(S)}{\partial t_n} (\delta(t_n - t_{n+1}) - \delta(t_n - t_{n-1})) + \frac{\partial^2 \log p_\theta(S)}{\partial t_n^2} \right] + C_1. \tag{15}$$

where the constant $C_1$ does not depend on $\theta$ and can be discarded. To construct the final empirical loss, we replace the expectation by the empirical average, which eliminates the delta functions as any two timestamps cannot overlap with each other, and obtain the final version:

$$\hat{F}(\theta) = \frac{1}{M} \sum_{m=1}^{M} \sum_{n=1}^{N_m} \frac{1}{2} \left( \frac{\partial \log p_\theta(S_m)}{\partial t_{m,n}} \right)^2 + \frac{\partial^2 \log p_\theta(S_m)}{\partial t_{m,n}^2} + C_1, \tag{16}$$

where we take $M$ sequences $\{S_m\}_{m=1}^{M}$ from $p(S)$, $t_{m,n}$ is the $n$-th timestamp on the $m$-th sequence.

It is worth noting that Sahani et al. [26] assumed the parametric density satisfies the smoothness property: $\partial_{t_n} \log p_\theta(S)|_{t_n = t_{n+1}} = \partial_{t_{n+1}} \log p_\theta(S)|_{t_{n+1} = t_n}$ to cancel most delta functions. Here, we emphasize that this smoothness assumption is not necessary. In our derivation, we do not utilize this smoothness property, but just take advantage of the non-overlapping of timestamps to eliminate all delta functions and obtain the same objective.

## A.2 Spatial Point Processes

Let us consider a planar point process for example. Given a 2-D observation region $\mathcal{X} \subseteq \mathcal{R}^2$, a realization from a 2-D spatial point process is composed of a random number of points $S = \{\mathbf{x}_n\}_{n=1}^N$ where $\mathbf{x}_n \in \mathcal{X}$ is the $n$-th event location (2-D coordinate). It is worth noting that these points are in no order. Similarly, we define a Fisher-divergence objective:

$$F(\theta) = \mathbb{E}_{p(S)} \frac{1}{2} \sum_{n=1}^{2N} \left( \frac{\partial \log p(S)}{\partial x_n} - \frac{\partial \log p_\theta(S)}{\partial x_n} \right)^2, \tag{17}$$

where $x_n$ is any entry in vector $\mathbf{x}_n$, i.e., $x_n \in \mathbf{x}_n$.

Similarly, we use the trick of integration by parts to eliminate the dependence of the loss on the gradient of the unknown ground-truth data distribution. Let us expand Eq. (17), discard $\left(\frac{\partial \log p(S)}{\partial x_n}\right)^2$ which does not depend on parameter $\theta$, and examine the cross-term:

$$\begin{aligned}
&\mathbb{E}_{p(S)} \left[ \frac{\partial \log p(S)}{\partial x_n} \frac{\partial \log p_\theta(S)}{\partial x_n} \right] \\
&= \int p(S) \frac{\partial \log p(S)}{\partial x_n} \frac{\partial \log p_\theta(S)}{\partial x_n} dS \\
&= \int_{S_{x_n^-}} \int_{x_n} \frac{\partial p(S)}{\partial x_n} \frac{\partial \log p_\theta(S)}{\partial x_n} dx_n dS_{x_n^-} \\
&= \int_{S_{x_n^-}} p(S) \frac{\partial \log p_\theta(S)}{\partial x_n} \Big|_{x_n=-\infty}^{x_n=+\infty} - \int_{x_n} p(S) \frac{\partial^2 \log p_\theta(S)}{\partial x_n^2} dx_n dS_{x_n^-} \\
&= \mathbb{E}_{p(S)} \left[ -\frac{\partial^2 \log p_\theta(S)}{\partial x_n^2} \right].
\end{aligned} \tag{18}$$

where $S_{x_n^-}$ represents the realization excluding $x_n$, the fourth line uses integration by parts, the fifth line assumes a weak regularity condition: $p(S)\partial_{x_n} \log p_\theta(S)$ goes to zero for any $\theta$ when $|x_n| \to \infty$. It is worth noting that in spatial point processes the limits of $x_n$ are no longer constrained because there is no order for the points. Therefore, the loss in Eq. (17) can be rewritten as:

$$F(\theta) = \mathbb{E}_{p(S)} \left[ \sum_{n=1}^{2N} \frac{1}{2} \left( \frac{\partial \log p_\theta(S)}{\partial x_n} \right)^2 + \frac{\partial^2 \log p_\theta(S)}{\partial x_n^2} \right] + C_1. \tag{19}$$

where the constant $C_1$ does not depend on $\theta$ and can be discarded. Replacing the expectation by the empirical average, we obtain the final empirical loss:

$$\hat{F}(\theta) = \frac{1}{M} \sum_{m=1}^M \sum_{n=1}^{2N_m} \frac{1}{2} \left( \frac{\partial \log p_\theta(S_m)}{\partial x_{m,n}} \right)^2 + \frac{\partial^2 \log p_\theta(S_m)}{\partial x_{m,n}^2} + C_1, \tag{20}$$

where we take $M$ realizations $\{S_m\}_{m=1}^M$ from $p(S)$, $x_{m,n}$ is the $n$-th variable on the $m$-th realization.

## A.3 Spatio-temporal Point Processes

It is interesting to see that both score matching estimators for temporal point processes and spatial point processes have the same empirical loss (see Eq. (16) and Eq. (20)) regardless of whether the points are sequential or not. Therefore, it is easy to draw the conclusion that for spatio-temporal point processes the empirical score matching estimator is:

$$\hat{F}(\theta) = \frac{1}{M} \sum_{m=1}^M \sum_{n=1}^{\tilde{N}_m} \frac{1}{2} \left( \frac{\partial \log p_\theta(S_m)}{\partial s_{m,n}} \right)^2 + \frac{\partial^2 \log p_\theta(S_m)}{\partial s_{m,n}^2} + C_1, \tag{21}$$

where $s_{m,n}$ is the $n$-th variable on the $m$-th sequence, $\tilde{N}_m$ is the number of equivalent variables on the $m$-th sequence, $\tilde{N}_m = N_m$ for 1-D point process, e.g., temporal point process; $\tilde{N}_m = 2N_m$ for 2-D point process, e.g., spatial point process; and $\tilde{N}_m = 3N_m$ for 3-D point process, e.g., spatio-temporal point process, etc.

## B  Denoising Score Matching Estimator

In Appendix A, we provide an estimator trying to match the gradient of the log-density of the point process model to the log-density of the point process data. However, the estimator requires the second derivatives, which is computationally expensive. To avoid this issue, following the derivation of Vincent [32], we derive a denoising score matching estimator.

Differently, the denoising score matching estimator tries to match the gradient of the log-density of the model to the log-density of the noisy point process data. We add a small noise to the sequence $S$ to obtain a noisy sequence $\tilde{S}$ (we add noise to each variable $\tilde{s}_n = s_n + \epsilon$), which is distributed as $p(\tilde{S}) = \int p(\tilde{S} \mid S)p(S)dS$. Therefore, the Fisher divergence between the noisy data distribution $p(\tilde{S})$ and model distribution $p_\theta(\tilde{S})$ is:

$$F_{\text{DSM}}(\theta) = \mathbb{E}_{p(\tilde{S})} \frac{1}{2} \sum_{n=1}^{\tilde{N}} \left( \frac{\partial \log p(\tilde{S})}{\partial \tilde{s}_n} - \frac{\partial \log p_\theta(\tilde{S})}{\partial \tilde{s}_n} \right)^2, \tag{22}$$

where $\tilde{s}_n$ is any entry in the noisy vector $\tilde{\mathbf{s}}_n$, i.e., $\tilde{s}_n \in \tilde{\mathbf{s}}_n = (\tilde{t}_n, \tilde{\mathbf{x}}_n)$, $\tilde{N}$ is the number of equivalent variables. Let us expand Eq. (22), discard $(\frac{\partial \log p(\tilde{S})}{\partial \tilde{s}_n})^2$ which does not depend on parameter $\theta$, and examine the cross-term:

$$\begin{aligned}
&\mathbb{E}_{p(\tilde{S})} \left[ \frac{\partial \log p(\tilde{S})}{\partial \tilde{s}_n} \frac{\partial \log p_\theta(\tilde{S})}{\partial \tilde{s}_n} \right] \\
&= \int p(\tilde{S}) \frac{\partial \log p(\tilde{S})}{\partial \tilde{s}_n} \frac{\partial \log p_\theta(\tilde{S})}{\partial \tilde{s}_n} d\tilde{S} \\
&= \int \frac{\partial}{\partial \tilde{s}_n} \int p(\tilde{S} \mid S)p(S)dS \frac{\partial \log p_\theta(\tilde{S})}{\partial \tilde{s}_n} d\tilde{S} \\
&= \iint p(S) \frac{\partial p(\tilde{S} \mid S)}{\partial \tilde{s}_n} \frac{\partial \log p_\theta(\tilde{S})}{\partial \tilde{s}_n} dSd\tilde{S} \\
&= \iint p(S)p(\tilde{S} \mid S) \frac{\partial \log p(\tilde{S} \mid S)}{\partial \tilde{s}_n} \frac{\partial \log p_\theta(\tilde{S})}{\partial \tilde{s}_n} dSd\tilde{S} \\
&= \mathbb{E}_{p(S,\tilde{S})} \left[ \frac{\partial \log p(\tilde{S} \mid S)}{\partial \tilde{s}_n} \frac{\partial \log p_\theta(\tilde{S})}{\partial \tilde{s}_n} \right].
\end{aligned} \tag{23}$$

Therefore, the loss in Eq. (22) can be rewritten as:

$$\begin{aligned}
&F_{\text{DSM}}(\theta) \\
&= \sum_{n=1}^{\tilde{N}} \mathbb{E}_{p(\tilde{S})} \frac{1}{2} \left( \frac{\partial \log p_\theta(\tilde{S})}{\partial \tilde{s}_n} \right)^2 - \mathbb{E}_{p(S,\tilde{S})} \left[ \frac{\partial \log p(\tilde{S} \mid S)}{\partial \tilde{s}_n} \frac{\partial \log p_\theta(\tilde{S})}{\partial \tilde{s}_n} \right] + C \\
&= \mathbb{E}_{p(S,\tilde{S})} \sum_{n=1}^{\tilde{N}} \frac{1}{2} \left( \frac{\partial \log p_\theta(\tilde{S})}{\partial \tilde{s}_n} \right)^2 - \frac{\partial \log p(\tilde{S} \mid S)}{\partial \tilde{s}_n} \frac{\partial \log p_\theta(\tilde{S})}{\partial \tilde{s}_n} + C \\
&= \mathbb{E}_{p(S,\tilde{S})} \frac{1}{2} \sum_{n=1}^{\tilde{N}} \left( \frac{\partial \log p(\tilde{S} \mid S)}{\partial \tilde{s}_n} - \frac{\partial \log p_\theta(\tilde{S})}{\partial \tilde{s}_n} \right)^2 + C_2,
\end{aligned} \tag{24}$$

where the constants $C$ and $C_2$ do not depend on $\theta$ and can be discarded. Replacing the expectation by the empirical average, we obtain the final empirical loss:

$$\hat{F}_{\text{DSM}}(\theta) = \frac{1}{2M} \sum_{m=1}^{M} \sum_{n=1}^{\tilde{N}_m} \left( \frac{\partial \log p(\tilde{S}_m \mid S_m)}{\partial \tilde{s}_{m,n}} - \frac{\partial \log p_\theta(\tilde{S}_m)}{\partial \tilde{s}_{m,n}} \right)^2 + C_2, \tag{25}$$

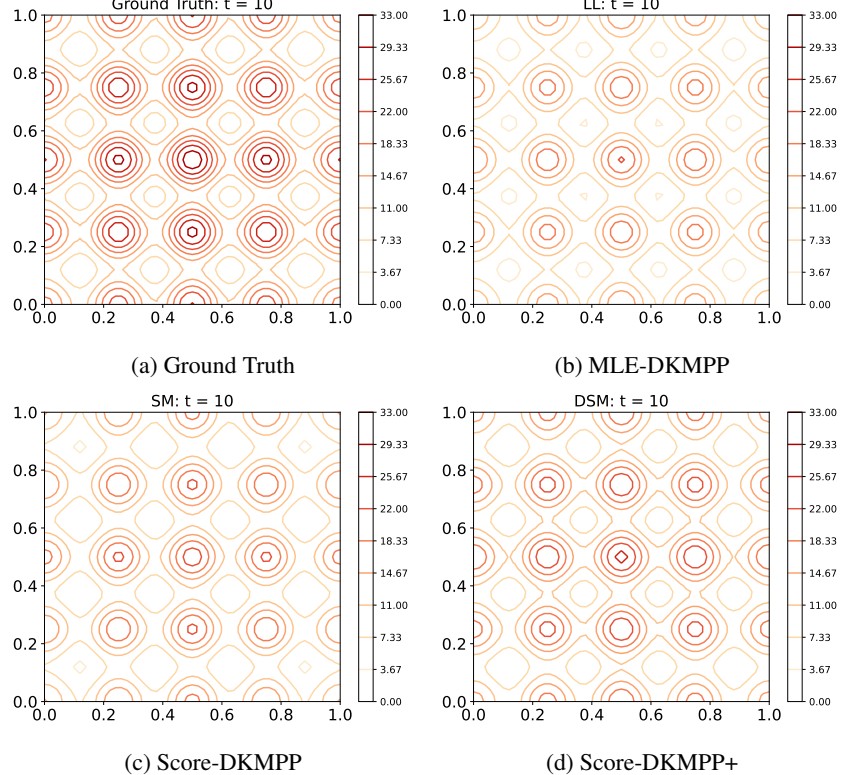

(a) Ground Truth

(b) MLE-DKMPP

(c) Score-DKMPP

(d) Score-DKMPP+

Figure 2: The intensity function at $t = 10$ estimated by three estimators (MLE-DKMPP, Score-DKMPP, Score-DKMPP+) with 1,000 Monte Carlo (MC) samples. Three estimators exhibit similar performance. Score-DKMPP and Score-DKMPP+ do not require Monte Carlo integration, and thus their estimation remain consistent regardless of MC samples. In contrast, MLE-DKMPP heavily relies on MC sampling and therefore its performance depends on the number of MC samples.

where we take $M$ clean and noisy sequences $\{S_m, \tilde{S}_m\}_{m=1}^M$ from $p(S, \tilde{S})$, $\tilde{s}_{m,n}$ is the $n$-th variable on the $m$-th noisy sequence, $\tilde{N}_m$ is the number of equivalent variables on the $m$-th noisy sequence.

## C Experimental Details

### C.1 Synthetic Data

**Data Simulation** We generate a 3-D spatio-temporal point process synthetic dataset. The spatial observation $\mathbf{x}$ spans the area of $[0, 1] \times [0, 1]$, while the temporal observation window covers the time interval of $[0, 10]$. We assume a 1-D covariate function $Z(\mathbf{u} = (\tau, \mathbf{r})) = (\mathcal{N}(r_1 \mid 0.5, 0.5) + \mathcal{N}(r_2 \mid 0.5, 0.5))$ on the domain. We set $f_{w_1}(\mathbf{u}_j, Z(\mathbf{u}_j)) = 20Z(\mathbf{u}_j) + 0.1$, $k_{\phi, w_2}(\mathbf{s}, \mathbf{u}_j) = k_\phi(g_{w_2}(\mathbf{s}), g_{w_2}(\mathbf{u}_j))$ where $k_\phi$ is the RBF kernel $k_\phi(\mathbf{x}, \mathbf{x}') = \exp(-\phi\|\mathbf{x} - \mathbf{x}'\|^2)$ with $\phi = 100$ and $g_{w_2}$ is a linear transformation $g_{w_2}(\mathbf{s}) = \mathbf{s} + 0.1$. We fix the representative points on a regular grid: 5 representative points evenly spaced on each axis, so there are $5^3 = 125$ representative points in total. We use the thinning algorithm to generate 5,000 sequences according to the intensity function specified above. The statistics of the synthetic data are shown in Table 3.

**Training Details** We fit a DKMPP model to the synthetic data with the ground-truth representative points and an RBF base kernel. Both the kernel mixture weight network $f$ and the non-linear transformation $g$ in the deep kernel are implemented using MLPs with ReLU activation functions. Therefore, the learnable parameters are $w_1, w_2, \phi$. The intensity functions at $t = 10$ estimated by three different estimators are shown in Fig. 2.

Table 3: The statistics of synthetic and real-world datasets.

| Dataset | Covariate Dimension | # of sequences | average # of events per sequence |
|---|---|---|---|
| Synthetic | 1 | 5,000 | 17 |
| Crimes in Vancouver | 1 | 1,096 | 87 |
| NYC Vehicle Collisions | 768 | 61 | 327 |
| NYC Complaint Data | 768 | 301 | 63 |

## C.2 Real-world Data

**Data Preprocessing** The preprocessing details of three real-world datasets are shown below. We listed the statistics of three real-world datasets after preprocessing in Table 3.

*Crimes in Vancouver* This dataset is composed of more than 530 thousand crime records, including all categories of crimes committed in Vancouver from 2003 to 2017. Each crime record contains the time and location (latitude and longitude) of the crime. We split the data into multiple sequences by year, month and day. Then, we select events from 2013 to 2016, drop the NaN value, and scale the time and space into a volume of $[0, 1] \times [0, 1] \times [0, 10]$. We select the categorical feature 'Crime Type' as the descriptive feature for each event. We convert the categorical feature into the corresponding numerical feature as the covariate. Finally, the dimension of covariate $\mathbf{Z}$ is 1.

*NYC Vehicle Collisions* The New York City vehicle collision dataset contains about 1.05 million vehicle collision records. Each collision record includes the time and location (latitude and longitude). We split the data into multiple sequences by day. We select the records from 01/01/2019 to 02/03/2019, drop the NaN value and scale the time and space into a volume of $[0, 1] \times [0, 1] \times [0, 10]$. We select 'BOROUGH', 'CONTRIBUTING FACTOR VEHICLE 1', 'CONTRIBUTING FACTOR VEHICLE 2' and 'VEHICLE TYPE CODE 1' as the descriptive features for each event. We concatenate several textual features and use a pre-trained DistilBERT [27] to extract the textual features, and concatenate the textual features with other numerical/categorical features as the covariate. Finally, the dimension of covariate $\mathbf{Z}$ is 768.

*NYC Complaint Data* This dataset contains over 228 thousand complaint records in New York City. Each record includes the date, time, and location (latitude and longitude) of the complaint. We split the data into multiple sequences by hour. We select the records from 01/11/2022 to 13/11/2022, drop the NaN value and scale the time and space into a volume of $[0, 1] \times [0, 1] \times [0, 10]$. We select 'OFNS_DESC', 'JURIS_DESC', 'LAW_CAT_CD', 'PD_DESC', 'VIC_RACE', 'VIC_SEX' and 'PREM_TYP_DESC' as the descriptive features for each event. We concatenate several textual features and use a pre-trained DistilBERT [27] to extract the textual features, and concatenate the textual features with other numerical/categorical features as the covariate. Finally, the dimension of covariate $\mathbf{Z}$ is 768.

**Training Details** Each dataset is divided into training, validation and test data using a $50\%/40\%/10\%$ split ratio based on time. For the real-world data, we fix the representative points on a regular grid: 5 representative points evenly spaced on each axis, so there are $5^3 = 125$ representative points in total. We use three different kernel functions for comparisons: the RBF kernel $k_\phi(\mathbf{x}, \mathbf{x}') = \exp\left(-\phi\|\mathbf{x} - \mathbf{x}'\|^2\right)$, the rational quadratic (RQ) kernel $k_\phi(\mathbf{x}, \mathbf{x}') = (1+\phi\|\mathbf{x}-\mathbf{x}'\|^2)^{-\frac{1}{2}}$, and the Ornstein-Uhlenbeck (OU) kernel $k_\phi(\mathbf{x}, \mathbf{x}') = \exp\left(-\phi\|\mathbf{x} - \mathbf{x}'\|\right)$. For the three real-world datasets, the kernel mixture weight network $f$ and the non-linear transformation $g$ in the deep kernel are implemented using MLPs with ReLU activation functions and we fixed the number of layers in $f$ and $g$ as 2.

**Hyperparameters** We tested the performance of Score-DKMPP and Score-DKMPP+ with different hyperparameters on three real-world datasets. We tested the number of representation points and network structures. For the representation points, we tested three different values, 64, 125, and 216, respectively. For the network structure, we tested the number of layers of 1, 2, and 4. When we tested the effect of representation points, we fix the network with 2 hidden layers and when we tested the effect of network structure, we fix the number of representation points as 125.

For the representation points, the accuracy of Score-DKMPP and Score-DKMPP+ overall have better performance when the number of representation points is 125. The accuracy tends to increase when

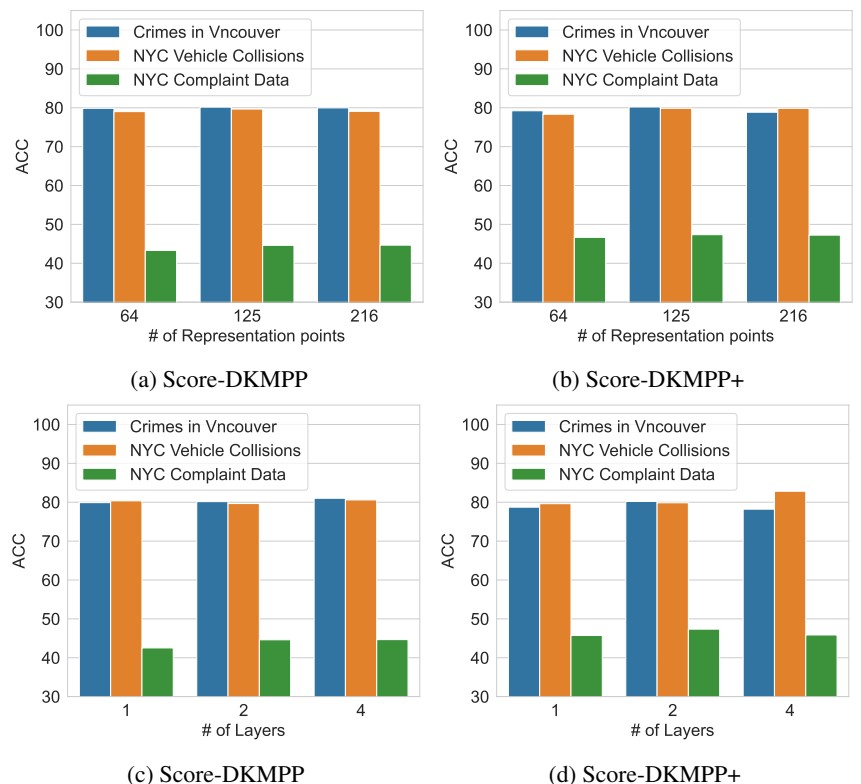

(a) Score-DKMPP  (b) Score-DKMPP+

(c) Score-DKMPP  (d) Score-DKMPP+

Figure 3: (a) The ACC performance of Score-DKMPP with the number of representation points of 64, 125 and 216; (b) the ACC performance of Score-DKMPP+ with the number of representation points of 64, 125 and 216; (c) the ACC performance of Score-DKMPP with the number of layers of 1, 2 and 4; (d) the ACC performance of Score-DKMPP+ with the number of layers of 1, 2 and 4.

the number of representation points increases from 64 to 125, however, except for the performance of Score-DKMPP on the complaint dataset, for other real-world datasets, the accuracy starts to drop.

For the network structure, the accuracy using Score-DKMPP on the Crimes in Vancouver and NYC Complaint data tends to increase as the number of layers increases. However, for Score-DKMPP+, we only capture a similar trend on the NYC Vehicle Collisions data. Other than those mentioned above, we do not observe a significant pattern when increasing the number of layers.

