# OpenReview forum: "Integration-free Training for Spatio-temporal Multimodal Covariate Deep Kernel Point Processes"
_NeurIPS.cc/2023/Conference — NeurIPS 2023 poster_

### Official Review · Reviewer_NRcd · 2023-06-18

**Soundness:** 3 good
**Presentation:** 4 excellent
**Contribution:** 3 good
**Rating:** 7
**Confidence:** 4

**Summary:**

This work proposes to fuse Deep Kernel Learning (DKL) into the Deep Mixture Point Processes (DMPP), resulting in Deep Kernel Mixture Point Processes (DKMPP) which can handle complex relationships between events and covariates in a more flexible and expressive manner. The authors also leverage the denoising score matching technique and come up with a training procedure that does not require integration and computation of second-order derivative. The proposed method is evaluated on a variety of datasets and compared with other baseline point process models to demonstrate its advantages.

**Strengths:**

• The authors extend the score matching to point processes and make the training more scalable by leveraging denoising technique, which, I believe, is a fair contribution to the field of point process learning and can be applied to a variety of scenarios.

• The authors conduct a wide range of experiments using both synthetic and real-world datasets, demonstrating the efficiency and effectiveness of the proposed method. The choice of model hyperparameters is appropriately studied.

• The organization and the presentation of the proposed method are clear and easy to understand.

**Weaknesses:**

On lines 158-159, the authors argue that Euclidean distance in the kernel may not be a suitable measure of similarity, especially for **high-dimensional inputs**. I think this statement needs more clarification. If I understand correctly, the input $\textbf{s}$ to the kernel function $k_{\phi}$ in the original DMPP lies in the space $\mathcal{R} \times \mathcal{R}^2$, but in DKMPP, based on (7), the input to the kernel function is first mapped by a deep neural network $g_{w_2}: \mathcal{R} \times \mathcal{R}^2 \rightarrow \mathcal{R}^D$ where $D$ can be much larger than the original dimension of $\textbf{s}$. In other words, the **high-dimension** of the input is induced by our design choice of $g_{w_2}$ instead of the data, is that right? Can $\textbf{s}$ itself be high-dimensional in spatio-temporal point processes?

Minor comments:

•	Since $\lambda(\textbf{s}|\mathcal{D})$ in (6) and (7) is an approximation of the true intensity of DMPP, I recommend using a different notation such as $\hat{\lambda}(\textbf{s}|\mathcal{D})$.

•	Figures 1(c)-(d) are displayed but never referred to. I suggest the authors either remove the figures or compare 1(c)-(d) to the true intensity function.


**Questions:**

• If $f_w$ is a network that outputs nonnegative mixture weights and $k_{\phi}$ is positive semi-definite, why do we need the link function $\eta(\cdot)$ to ensure non-negativity in (7) but not in (6)?

• Why do all point process models yield relatively low accuracy on the NYC Complaint data? Does this mean point process models are not good at processing textual information?


**Limitations:**

The limitations are adequately addressed. I’m not aware of any direct negative societal impacts of this work.

---

> ### Author Rebuttal · Authors · 2023-08-06
>
> > Q1: "On lines 158-159, the authors argue that Euclidean distance in the kernel may not be a suitable measure of similarity, especially for high-dimensional inputs. I think this statement needs more clarification. If I understand correctly, the input $\mathbf{s}$ to the kernel function $k_\phi$ in the original DMPP lies in the space $\mathcal{R}\times\mathcal{R}^2$, but in DKMPP, based on (7), the input to the kernel function is first mapped by a deep neural network $g_{w_2}: \mathcal{R}\times\mathcal{R}^2\to\mathcal{R}^D$ where $D$ can be much larger than the original dimension of $\mathbf{s}$. In other words, the high-dimension of the input is induced by our design choice of $g_{w_2}$ instead of the data, is that right? Can $\mathbf{s}$ itself be high-dimensional in spatio-temporal point processes?"
>
> A: The original intention behind deep kernel [1] was to overcome the limitations of Euclidean distance and enhance the expressive power of the kernel. If the raw data, denoted as $\mathbf{x}$, has high dimensionality, [1] has shown that Euclidean distance is an unsuitable measure of similarity. If we first pass the raw data $\mathbf{x}$ through a deep neural network to obtain a feature (which may also be high-dimensional), the advantage lies in the deep kernel's ability to learn metrics by optimizing the input space transformation in a data-driven manner. Thus, we emphasize the significance of handling high-dimensional raw data $\mathbf{x}$ where Euclidean distance is unsuitable for measuring similarity.
>
> Regarding spatio-temporal point processes, we understand your concern, as the dimensionality of $\mathbf{s}$ seems not very high.
> However, in our experiments, we found that even for this three-dimensional problem, Euclidean distance is not optimal. The deep kernel's ability to learn metrics from data outperforms Euclidean distance. We will provide further clarification on this statement in the camera-ready version.
>
> [1] Wilson, A. G., Hu, Z., Salakhutdinov, R., \& Xing, E. P. (2016, May). Deep kernel learning. In Artificial intelligence and statistics (pp. 370-378). PMLR.
>
> > Q2: "Since $\lambda(\mathbf{s}|\mathcal{D})$ in (6) and (7) is an approximation of the true intensity of DMPP, I recommend using a different notation such as $\hat{\lambda}(\mathbf{s}|\mathcal{D})$."
>
> A: Thanks for your suggestion. We agree and will correct this in camera ready.
>
> > Q3: "Figures 1(c)-(d) are displayed but never referred to. I suggest the authors either remove the figures or compare 1(c)-(d) to the true intensity function."
>
> A: Figures 1(c)-(d) are referred to. Please see lines 344-350.
>
> > Q4: "If $f_w$ is a network that outputs nonnegative mixture weights and $k_\phi$ is positive semi-definite, why do we need the link function $\eta(\cdot)$ to ensure non-negativity in (7) but not in (6)?"
>
> A: In the original DMPP (Eq. (6)), $f_w$ is a deep neural network that outputs nonnegative mixture weights, so we do not need the link function. However, in our proposed DKMPP (Eq. (7)), we eliminate the constraint that the mixture weight $f_{w_{1}}$ must be non-negative. Therefore, to ensure the non-negativity of the intensity function, DKMPP introduces a link function $\eta(\cdot)$. Maybe we should write $\tilde{f}_{w_1}$ in Eq.(7) to distinguish it from $f_w$ in Equation (6) to avoid confusion. We will provide further clarification on this statement in the camera-ready version.
>
> > Q5: "Why do all point process models yield relatively low accuracy on the NYC Complaint data? Does this mean point process models are not good at processing textual information?"
>
> A: It seems that the low accuracy observed with point process models on the NYC Complaint dataset is attributed to the dataset's specific characteristics, rather than an inherent limitation of point process models in processing textual information. Because on the NYC Vehicle Collisions dataset, we also used textual covariate information and achieved very good accuracy.
> The overall low performance on the NYC Complaint dataset is due to the dataset's poor data quality. We found that all models' performance on the complaint dataset is poor, primarily due to the following reasons: the predictions are made within a relatively short period, and the number of events on each sequence fluctuates significantly. Consequently, even when our model's predicted values closely align with the average value of multiple sequences, the accuracy remains low for each individual sequence. However, even in this situation, our model still outperforms other baselines.

---

> > ### Comment · Reviewer_NRcd · 2023-08-16
> > **Response to author rebuttal**
> >
> > Thank you for answering my questions. The responses mostly make sense to me and I'm inclined to maintain the score as is.

---

### Official Review · Reviewer_6cnn · 2023-07-05

**Soundness:** 3 good
**Presentation:** 3 good
**Contribution:** 2 fair
**Rating:** 5
**Confidence:** 3

**Summary:**

This paper argues that there are two common approaches for modeling intensity functions: traditional and covariate based methods, and this paper focuses on the latter one. In detail, the intensity function is designed in a kernel convolution form: $\lambda(s|\mathcal{D})=\int f_w(\mathbf{u},\mathbf{Z(u)})k_{\phi}(\mathbf{s,\mathbf{u}})d\mathbf{u}$, in which contextual information is embedded in $\mathbf{Z(u)}$. In practice, its integration is replaced by summation. This method makes integration $\int \lambda(s|\mathcal{D})ds$ intractable. However, the kernels lack expressiveness both because of the unknown relationship between covariates and event occurrence and improper usage of Euclidean distance. To solve this issue, authors propose to use deep kernels which are modeled by neural networks. To address parameter estimation, the authors further propose to use of a score matching-based estimator to estimate parameters.

**Strengths:**

1. The theoretical part is reasonable and complete, and the proposed model effectively takes into account both solving the difficulty of integrating the intensity function and maintaining the strong expressiveness of the intensity function.

3. The score matching-based modeling method is novel and interesting and plays a positive role in promoting research in the field of point processes.

3. The advantages of the model are adequately and effectively demonstrated by experiments.

**Weaknesses:**

1. The motivation for using a score-based approach is not clear. In fact, the score-based approach is a special generation model. In that case, why not use another generation model, such as GAN or VAE? The authors point out that the score-based approach can effectively solve the parameter estimation problem, but it seems that other generative models can also solve the problem.

2. The proposed model seems to be a combination of existing frameworks, which actually hinders its nolvity to some extend.

**Questions:**

1. I'm confused about the reason why the authors use score-based methods.

2. Actually, there are some efforts concerning the embedding of generative models into point processes. Is there any comparison between the proposed model and some existing methods, see, for example [1-2].


[1] Xiao S, Farajtabar M, Ye X, et al. Wasserstein learning of deep generative point process models[J]. Advances in neural information processing systems, 2017, 30.
[2] Mehrasa N, Jyothi A A, Durand T, et al. A variational auto-encoder model for stochastic point processes[C]//Proceedings of the IEEE/CVF Conference on Computer Vision and Pattern Recognition. 2019: 3165-3174.

**Limitations:**

Yes, the authors adequately addressed the limitations and potential negative societal impact of their work.

---

> ### Author Rebuttal · Authors · 2023-08-06
>
> > Q1: "The motivation for using a score-based approach is not clear. In fact, the score-based approach is a special generation model. In that case, why not use another generation model......"
> "I'm confused about the reason why the authors use score-based methods."
> "Actually, there are some efforts concerning the embedding of generative models into point processes......"
>
> A: It appears that the reviewer may have confused score matching with generative models. While in recent years score matching has been used for training generative models such as diffusion models, score matching itself is originally an estimator used for estimating model parameters, rather than being a generative model.
> When score matching was originally invented [1], it was designed for estimating parameters of non-normalized statistical models, rather than for generative models. This is clearly indicated in the title of [1].
>
> Similarly, in our current work, we use score matching as an estimator for point process parameters, as point processes themselves can be understood as non-normalized models: the compensator in the log-likelihood (the second term in Eq. (8)) can be understood as an intractable normalizing constant.
> As indicated in lines 46-49, 53-56, 184-188 in our paper, the compensator in the log-likelihood is typically an intractable integral that usually requires numerical integration methods, leading to numerical errors and computational inefficiency. Therefore, we adopted score matching to avoid the computation of the compensator, hence the name "integration-free" for our paper.
>
> In conclusion, our focus is to find an alternative estimator to MLE that does not require integration, rather than designing a generative model for point processes.
>
> [1] Hyvärinen, Aapo, and Peter Dayan. "Estimation of non-normalized statistical models by score matching." Journal of Machine Learning Research 6.4 (2005).
>
> > Q2: "The proposed model seems to be a combination of existing frameworks, which actually hinders its nolvity to some extend."
>
> A: We politely disagree. All research builds upon existing works. As far as we know, few works have attempted to utilize score matching for the estimation of point processes. Our work appears to be the first attempt to apply score matching to covariate-based deep spatio-temporal point processes.

---

> > ### Comment · Reviewer_6cnn · 2023-08-12
> >
> > After re-reading the author's detailed response and the paper, I agree with the author's emphasis on the work, that is, the focus of the paper is on designing an alternative estimator to MLE rather than on proposing a generative model-based point processes.  To this end, I raised my score.

---

### Official Review · Reviewer_FYsq · 2023-07-07

**Soundness:** 3 good
**Presentation:** 3 good
**Contribution:** 2 fair
**Rating:** 6
**Confidence:** 3

**Summary:**

The paper proposes an enhanced version of Deep Mixture Point Processes with a flexible neural network-based kernel. The intractable training process of the point process with deep kernel is handled by a score-matching technique with the denoising method.

**Strengths:**

1. The proposed deep kernel goes beyond the parametric kernel and substantially improves the model flexibility and expressiveness.
2. Address the learning challenges of MLE (a long-lasting problem in neural pp training) by proposing a score-matching technique which achieves better modeling performance and computational efficiency. This is also supported by the experimental results.

**Weaknesses:**

1. Related works are not comprehensive enough. There is plenty of work on point processes equipped with deep kernel, such as Okawa [1] and Zhu. [2]. These works can be reviewed to make the paper more comprehensive.
2. More experiments can be included.  For example, one baseline of Hawkes process equipped with a deep kernel can be included. Also model performance on synthetic point process data (such as self-exciting point process or self-correcting process) would make the numerical results more convincing.

---
[1] Maya Okawa, Tomoharu Iwata, Yusuke Tanaka, Hiroyuki Toda, Takeshi Kurashima, and Hisashi Kashima. Dynamic hawkes processes for discovering time-evolving communities’ states behind diffusion processes. In Proceedings of the 27th ACM SIGKDD Conference on Knowledge Discovery & Data Mining, pages 1276–1286, 2021.
[2] Shixiang Zhu, Haoyun Wang, Zheng Dong, Xiuyuan Cheng, and Yao Xie. Neural spectral marked point processes. In International Conference on Learning Representations, 2022.

**Questions:**

1. How do we calculate the model density $p_\theta(\tilde{S}_m)$ in equation 12?
2. Could the authors include additional synthetic experiments, such as modeling self-exciting point process data, because such data is becoming ubiquitous and gaining popularity in recent research and real-world application.

**Limitations:**

The current structure still imposes parametric constraints on the kernel. An alternative form of the kernel can be considered [1].

---
[1] Shixiang Zhu, Haoyun Wang, Zheng Dong, Xiuyuan Cheng, and Yao Xie. Neural spectral marked point processes. In International Conference on Learning Representations, 2022.

---

> ### Author Rebuttal · Authors · 2023-08-06
>
> > Q1: "Related works are not comprehensive enough. There is plenty of work on point processes equipped with deep kernel, such as Okawa [1] and Zhu. [2]. These works can be reviewed to make the paper more comprehensive."
>
> A: Thank you for your suggestion. Because we cannot make changes to the manuscript during the rebuttal stage, we will review those works you mentioned in the camera-ready.
>
> > Q2: "How do we calculate the model density $p_{\theta}(\tilde{S}_m)$ in equation 12?"
>
> A: $p{\theta}(\tilde{S}_m)$ and $p{\theta}(S_m)$ share the same $p{\theta}(\cdot)$ that represents the probability density function (likelihood function) corresponding to the parameterized point process model. The only difference between the two lies in the sequences used: $p{\theta}(S_m)$ utilizes clean sequences, while $p{\theta}(\tilde{S}_m)$ in equation 12 utilizes noisy sequences.
>
> > Q3: "Could the authors include additional synthetic experiments, such as modeling self-exciting point process data, because such data is becoming ubiquitous and gaining popularity in recent research and real-world application."
>
> A: This is a misunderstanding. We would like to clarify that our proposed model is not a history-dependent point process model but rather a covariate-dependent point process model. In other words, we focus on the impact of covariates on point process dynamics, rather than the influence of past events on subsequent point process dynamics. However, we appreciate your valuable feedback, and in future work, we will consider the mutual influences among events, such as self-exciting point processes or self-correcting processes.

---

> > ### Comment · Reviewer_FYsq · 2023-08-15
> > **Reply to the author rebuttal**
> >
> > The authors' rebuttal appropriately addresses my concerns and questions. I believe the neural network-based kernel method and the score-matching technique would contribute to the point process community in the future. In light of this, I would raise my score.

---

### Author Rebuttal · Authors · 2023-08-06

We would like to express our gratitude to all the reviewers for their valuable efforts in providing insightful comments and constructive feedback. We are pleased that the reviewers have recognized the significance of our paper in solving an interesting covariate point process estimation problem, proposing efficient score-based estimators, conducting comprehensive numerical experiments, and maintaining clear and concise writing.

In the following, we address reviewers' comments point by point. We hope that our responses adequately address the concerns raised by the reviewers. Should any further doubts or questions arise, please do not hesitate to reply. Thank you once again for your time and effort in reviewing our work.

---

### Decision · Program_Chairs · 2023-09-21

**Decision:**

Accept (poster)

**Comment:**

This paper extends work on neural point process models by allowing the intensity function to be modeled by deep kernels. Additionally, the authors investigate score function matching as an alternative to Monte Carlo integration when computing log likelihoods. The score function-based likelihood - though briefly explored in prior work - is a notable contribution of this work that addresses a long-standing problem in non-analytic point process models. The authors missed several important related works that also use deep kernels for point process modeling, though they did promise to include comparisons to these works in the final revision. Nevertheless, the efficacy of the proposed method, coupled with the use of score matching-based inference, will be of interest to the NeurIPS community, and therefore I recommend acceptance.